# Multicomponent Transient Electromagnetic Exploration Technology and Its Application

Xingchun Wang [1,2,*], Qingquan Zhi [1,2,*], Junjie Wu [1,2], Xiaohong Deng [1,2], Yue Huang [1,2], Qi'an Yang [3] and Jinhai Wang [4]

1 Institute of Geophysical and Geochemical Exploration, Chinese Academy of Geological Sciences, Langfang 065000, China; wjj498211@126.com (J.W.); dengxiaohong@mail.cgs.gov.cn (X.D.); huangyue@mail.cgs.gov.cn (Y.H.)
2 Laboratory of Geophysical EM Probing Technologies, Ministry of Natural Resources, Langfang 065000, China
3 Qinghai No. 5 Geological and Mineral Exploration Institute, Xining 810013, China; wkyyqa@163.com
4 Qinghai No. 3 Geological and Mineral Exploration Institute, Xining 810008, China; huzi115@163.com
* Correspondence: wxingchun@mail.cgs.gov.cn (X.W.); zhiqingquan@gmail.com (Q.Z.)

**Abstract:** To take full advantage of the multicomponent transient electromagnetic method, we summarize the advantages of multicomponent exploration based on simulation calculation. We have carried out experiments on the effectiveness of the method in the known copper-nickel mining area. The results show that the characteristic curve of the horizontal component can effectively point to the central direction of the low-resistivity ore body and reflect the occurrence of the ore body. The degree of coupling between the vertical component and the ore body is high, which is beneficial for quantitative inversion. The results for the horizontal component and vertical component interpretations are consistent, which reduces the limitation of conventional single vertical component interpretation and effectively improves the work efficiency in field work.

**Keywords:** transient electromagnetic method; multicomponent; zero contour; directivity; occurrence; copper-nickel deposit

## 1. Introduction

In recent decades, the vertical component of the transient electromagnetic method (TDEM) has been a research focus because of its easy measurement, high signal-to-noise ratio, easy inversion, and easy interpretation [1–7]. However, the horizontal-component (HC) signal is weak, and the signal-to-noise ratio is low, which makes its interpretation relatively slow. With the emergence of a three-component (3C) TDEM system, research on 3C interpretation technology has become popular, including ground and borehole TDEM [8–15]. Some scholars have simulated the multicomponent aspects of the fixed loop, analyzed HC characteristics with various occurrences of plate, and made interpretations in combination with the geological section. These studies have shown that the HC information is helpful for interpreting anomalies; however, the researchers only made single-profile analyses [16,17]. Jintao Liu has analyzed the advantages of multicomponent interpretation and has obtained good results from the field data measured in a hydrogeological survey [18]. Zihao Han has calculated the different components of TDEM. The results show that the HC is more sensitive to the low-resistivity body than the verticals; however, the author provided only predictive suggestions without further analysis [19]. Shaocong Tan has illustrated the superiority of the multicomponent interpretation by defining the characteristic function of the ratio vertically and horizontally [20]. By changing the relative position between the transmitter loop and the plate, Wang has carried out a 3C simulation of the fixed loop. The results show that the combination of HC information is helpful in determining the ground projection center and the attitude of the plate [21]. Zonge and Carlson have discussed the role of multiple components in pipeline detection, and their research results have shown

that HC can identify the cables underground that vertical components cannot [22,23]. Chen has made a multicomponent simulation, and the results show that multicomponent data is superior to a single component in the judgment of conductor direction and position, especially in the detection of unknown explosives [24].

By changing the size and orientation of the plate in the homogeneous half-space, we carried out 3C forward simulation of fixed loop and found that the variation characteristics of the HC zero contour were closely related to the orientation and center of the plate. For fixed-loop devices, we know that a rectangular loop has two symmetric axes, and we generally define the direction of the X component parallel to the field line, so thatthe zero contour of the X component is approximately perpendicular to the field line.

The HC profile of a fixed loop always has a zero point, when the medium subsurface is a homogeneous half-space or layered medium. The zero-value point is on the axis of symmetry; otherwise, the zero-value point of the X component will deviate from the axis of symmetry of the transmitting loop. As shown in Figure 1, the size of the transmitting loop is 500 m × 900 m, and 5 lines are arranged symmetrically in the loop with a distance of 100 m; the length of line is 800 m. The line is L0, L100, L200, L300, and L400 from south to north. L200 coincides with the east-west symmetry axis of the transmitting loop.

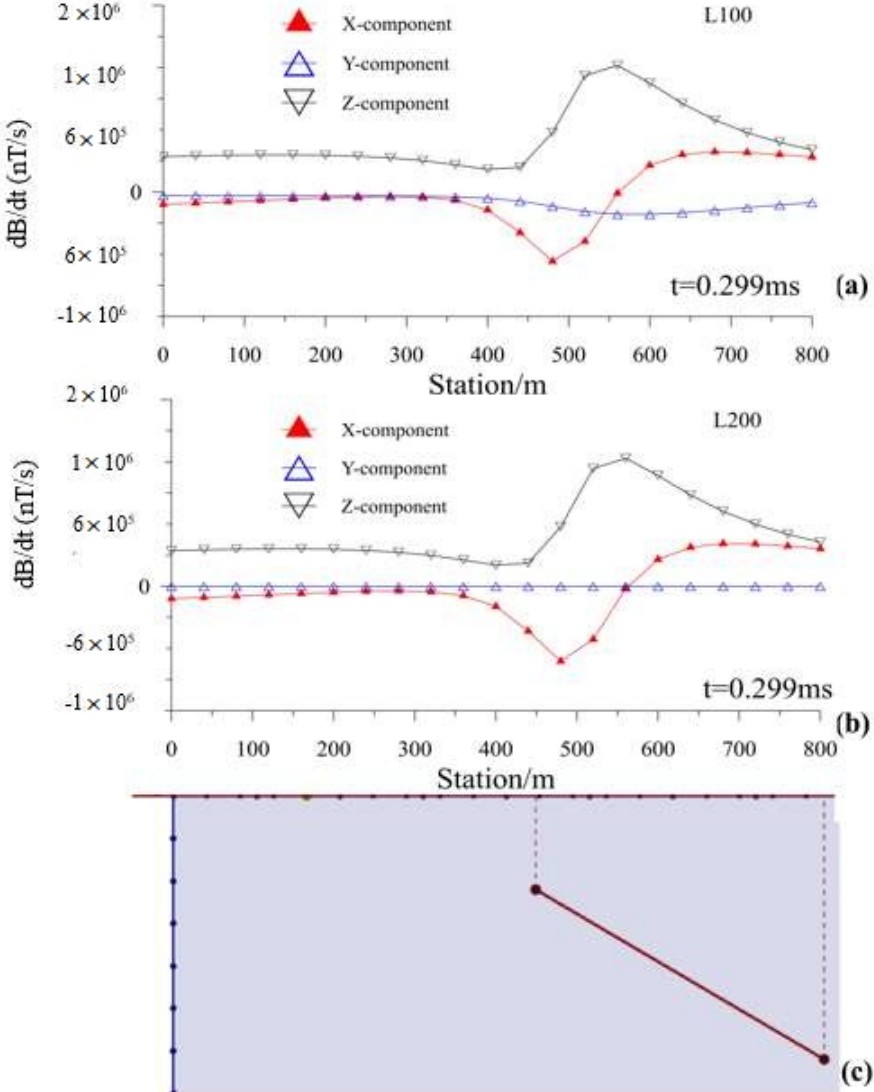

**Figure 1.** Characteristic curves of 3C TDEM simulation on profile L100 (**a**) and profile L200 (**b**) at sampling time 0.299 ms with the forward resistivity model (**c**).

Figure 1a,b exhibit the 3C profiles for L100 and L200, respectively. Figure 1c shows the forward resistivity model. It is a homogeneous half-space with resistivity of 800 $\Omega\cdot$m. The size of the plate inside the model is 400 m $\times$ 400 m, with conductivity of 20 S. The coordinates of the top center of the plate are (480, 200, $-$65), and the plate inclines 30 degrees toward the east. As shown in Figure 1a,b, the Z-components show apparently positive features, while the X-components cross the zeros. The Y-components show negative anomalies for profile L100 but show near zeros for L200 because of the symmetry.

When there are many lines in the loop, the zero-value points of all the lines form a curve, which is called the zero-contour line, and it coincides with the axis of symmetry when subsurface medium is a homogeneous half-space or layered medium [25].

In this paper, we performed multicomponent-3D-forward modeling of the fixed loop for plates with different sizes (relative to the transmitting loop) and summarize its rules. We then made an analysis of the multicomponent TDEM data collected from a copper-nickel ore deposit. The results show that multicomponent data analysis is superior to single-component data in determining the central location and occurrence of ore bodies. In addition, the qualitative interpretation of the HC is consistent with the quantitative inversion of the vertical component.

## 2. Forward Modelling

### 2.1. Multicomponent Response Characteristics of a Small, Inclined plate

When the medium underground does not satisfy the one-dimensional assumption, the HC is no longer equal to zero, and the central and occurrence of the low-resistivity body subsurface can be inferred. Figure 2a shows a 3D forward model of an eastward-inclined plate in a homogeneous half-space. The parameters were as follows: line spacing 100 m in the east-west direction, field line length 900 m, station spacing 50 m, Crone 50 ms standard sampling channel selected, ramp 1 ms, transmitter loop size 1000 m $\times$ 1000 m, transmitter current 20 A, homogeneous half-space resistivity was 600 $\Omega\cdot$m, plate size 400 m $\times$ 200 m, plate conductivity 100 S, buried plate depth 175 m, and plate incline to the east 35 degrees. We received the data in a loop, and the stations started from point 50 and ended at 950 (Figure 2b). The large red loop was the transmitter loop, and the inner light green loop was the projection of the inclined plate on the ground.

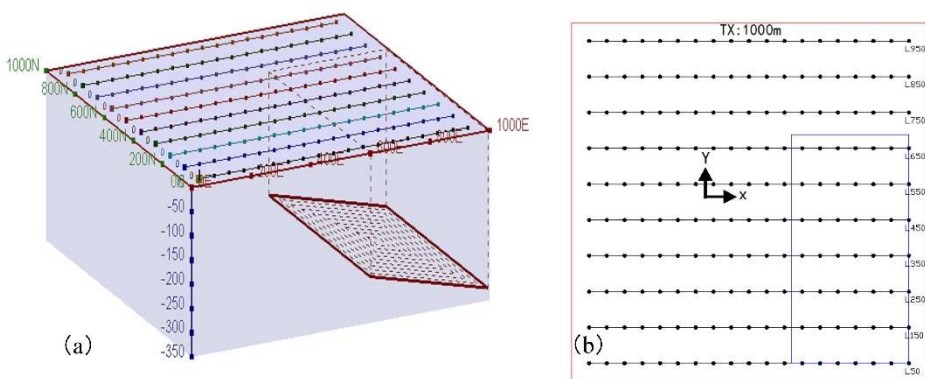

**Figure 2.** 3D forward model of the fixed loop: (**a**) 3D forward model, (**b**) plan for 3D model.

Figure 3 shows the relationship between the Z-component contour and the HC zero contour with different delays. In Figure 3, the black dotted line is the X component's zero contour, the blue line is the Y component's zero contour, the background is the Z-component's contour map, and the red, dotted rectangle is the projection of the plate. With sampling time increasing, the abnormal center of the Z component gradually approached the center of the surface projection until they coincided. We also found that in the early period, the position of the HC's zero contour was close to the axis of symmetry for the transmitting loop. With increasing sampling time, the HT's zero contour gradually deviated from the axis of symmetry due to the existence of the plate, and their intersection gradually

moved closer to the center of the surface projection. The variation trend of the characteristic curve of the HC and the anomaly center of the Z component pointed to the anomaly center. This indicated that the HC had directivity, but we could not judge whether the plate was inclined or not.

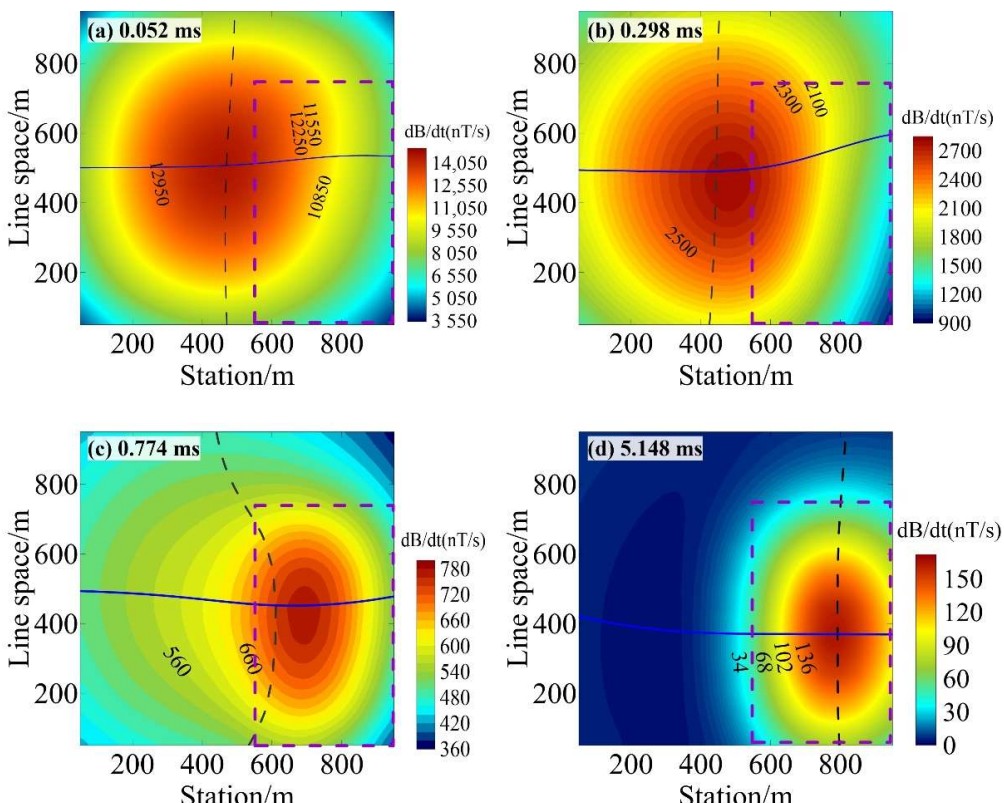

**Figure 3.** Three-component feature synthesis map with different sampling times (inclined plate): (**a**) 0.052 ms; (**b**) 0.298 ms; (**c**) 0.774 ms; (**d**) 5.148 ms.

### 2.2. Multicomponent Response Characteristics of a Big, Inclined Plate

In field work, we rarely know the distribution characteristics of the ore body; the transmitter loop and the ore body cannot be well coupled. Therefore, we designed the model shown in Figure 4 to illustrate the advantages of HC.

The model parameters are as follows: the size of transmitter loop is 800 m × 900 m, and eight east-west lines are arranged symmetrically in the loop; line space is 100 m; and its length is 800 m. It is a homogeneous half-space with resistivity of 1000 Ω·m. The size of the plate inside the model is 300 m × 800 m, with conductivity of 30 S. The coordinates of the top center of the plate are (−560, 355, −40). It inclines 25 degrees toward the east.

Figure 4a is a 3D diagram, and Figure 3b is a plane-relative position diagram of the plate and the zero contour of the X-component with different delays.

As shown in Figure 3b, the dashed quadrilateral is the projection of the plate on the ground; with increasing sampling time, the X-component zero contour gradually moves westward and points to the projection center of the plate. We have completed the simulation for X component, and this characteristic also applies to the Y component, and their intersection must point to the projection center of the plate. Figure 5 is the Z-component contour map with different sampling delays. With increasing sampling time, the anomaly center of the Z component gradually approached the projection center of the plate.

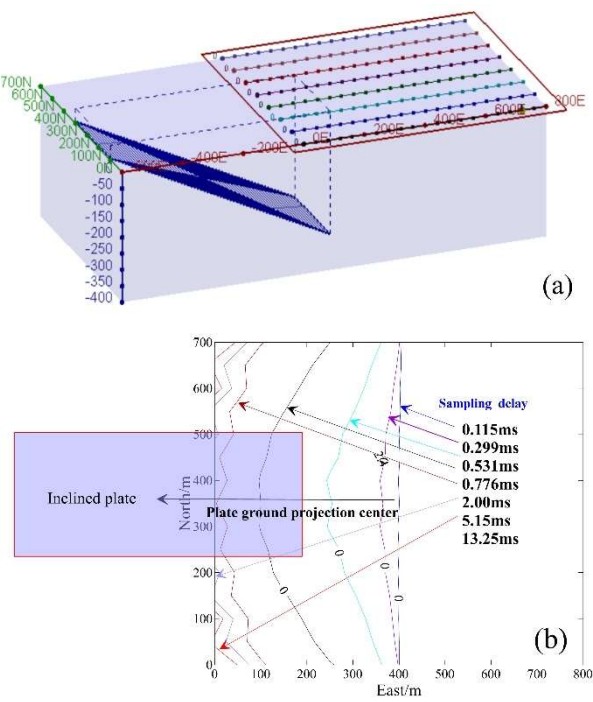

**Figure 4.** 3D forward models for fixed-loop and X-component characteristic curves: (**a**) 3D forward model; (**b**) relative position of the plate and the X-component zero contour.

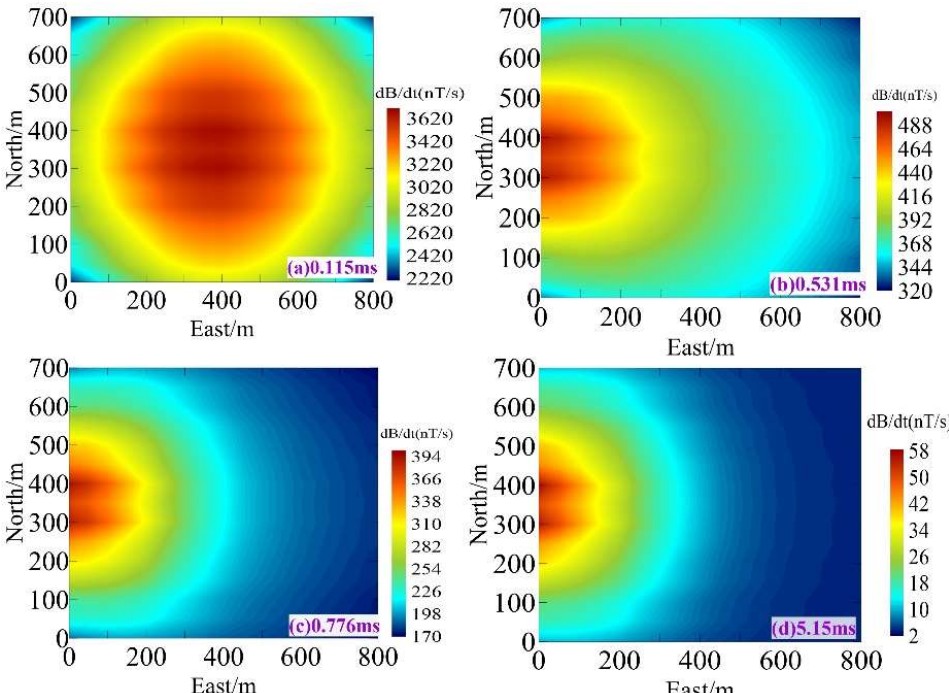

**Figure 5.** Three-component feature synthesis maps with different sampling times: (**a**) 0.115 ms; (**b**) 0.531 ms; (**c**) 0.776 ms; (**d**) 5.15 ms.

Combining the results of Figures 4b and 5, we find that the HC's zero contour helps determine the low-resistance ore body's center. The zero contour of the X component moved west, contrary to the eastward dip direction of the plate set in the previous. We can infer the plate's inclination according to the variations in the HC zero contour. It also shows that after the ore-body positioning is determined by 3C TDEM, the later work can

be carried out in a targeted manner, which is conducive to improving the efficiency of field work.

### 2.3. Analysis of Forward Modeling Results

The functions of the X component and the Y component are equivalent. That the direction of the survey line is the same as that of the X component is a constructed factor. According to the relative size and inclination angle of the plate and the coupling degree between the transmitter loop and the plate, we have completed the corresponding forward calculation. It turned out that the changes in parameters, such as the relative size, inclination angle, and position of the subsurface plate relative to the transmitting loop led to changes in the position of the HC zero contour, and the corresponding vertical-component-plane-anomaly center also changed. Interpretation of multicomponent TDEM data can predict the dips and center positions of ore bodies underground.

## 3. Multicomponent Transient Electromagnetic Method Tests in Copper-Nickel Ore

This copper-nickel deposit is found in the hinterland of east Kunlun with a higher degree of exploration. TDEM was listed as one of the main methods according to the requirements and objectives of the validity test of geophysical methods. According to the orebody distribution characteristics, we completed forward modeling and finally selected the fixed-source 3C measurement method.

### 3.1. Geological Setting

The survey area is on the northern slope of the western section of east Kunlun and the southern margin of the Qaidam Basin. The tectonic units belong to the east Kunlun arc basin system of the Qin-Qi-Kun orogenic system (Figure 6). The tectonic pattern of the arc basin system is relatively clear, and the main orogenic period is Caledonian [26].

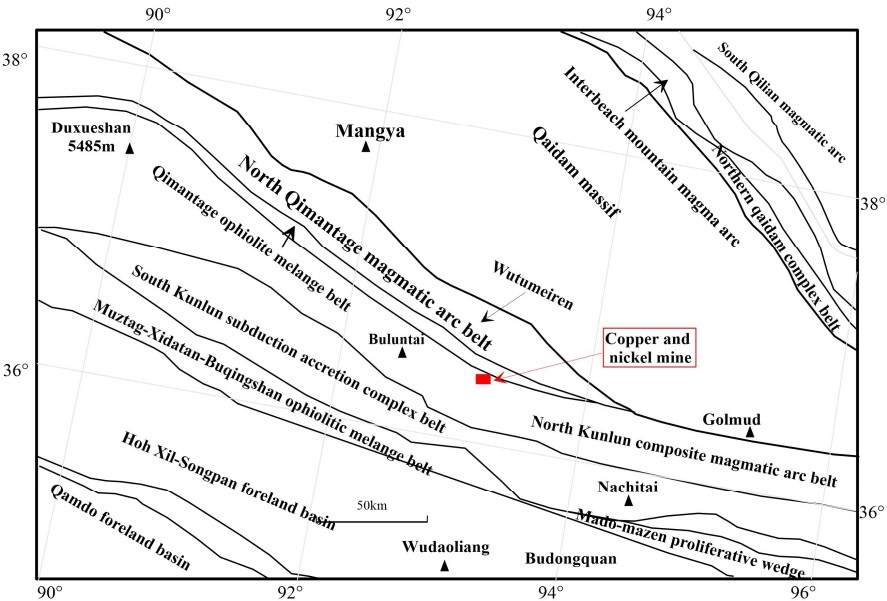

**Figure 6.** Regional structure map of the Cu-Ni mining study area.

The deposit is a magmatic molten copper-nickel sulfide deposit. The nickel ore occurs in peridotite and pyroxenite. It is hidden under the Jinshuikou group in the section. The rock comprises pyroxenite, peridotite, oligoclase, and gabbro. The ore-bearing lithology is mainly dihedral peridotite and pyroxenite. The ore minerals are mainly chalcopyrite, nickel pyrite, magnetite, and pyrrhotite [27,28].

Most ore bodies occur in layered form, and the upper part is mainly disseminated and massive ore, while the middle and lower part are mainly densely disseminated and dense

massive ore. A few ore bodies are lenticular and funnel-shaped in the upper part of the rock mass, forming overhanging ore bodies, or are banded in the rock mass.

The measurement results of the physical properties of rocks show that the ore rock in the area has the characteristics of "low resistivity, high polarization, high magnetism, and high density." In this area, gravity, magnetic, electromagnetic, and other geophysical work have been carried out, effectively delineating the boundaries of ore-bearing rock mass. On the geophysical characteristic parameters of the surrounding rock and ore bearing in the Cu-Ni mining area, we found that the profiles' distribution characteristics of resistivity, polarization, and magnetic susceptibility objectively reflected the occurrence characteristics of ore bodies [29,30].

### 3.2. Method and Technique

Due to the steep terrain, we selected a fixed loop for data acquisition and the Pulse-EM system produced by the Crone Company of Canada for receiving and transmitting. The system is equipped with a rod core probe that can realize HC data acquisition when adjusted to horizontal state; the sensor is 58 mm in length and 63 mm in diameter, and its effective area is 3850 square meters.

The geologic profile was in a north-south direction with a drilling distance of 80 m. The direction of the three components followed the right-hand rule, in which the positive direction of the X component is consistent with the northwest direction of the survey line; the positive direction of the Y component is the southwest direction, perpendicular to the survey line; and the Z component is vertically upward. According to terrain conditions and other factors, we arranged a 600 m × 600 m transmitting loop on the ground, as shown in Figure 7. The green rectangle is the transmitting loop. According to the field test results and the geological profile position, the main parameters were as follows: the transmitting current was 15 A, the time base was 50 ms, the line distance was 80 m, the station distance was 50 m, and the 3C survey lines from east to west were Lines 9, 11, 13, 15, and 17. Due to the steep terrain, we abandoned some points during collection.

All the red exploration points in Figure 7 are ore-seeing boreholes. According to the differences in ore body occurrence, we chose to lay the transmitting loop on the north side of exploration Lines 9–17. We collected the data of 3C inside and outside the transmitting loop. Due to the limitations of interpretation technology, we only studied the data inside the transmitting loop.

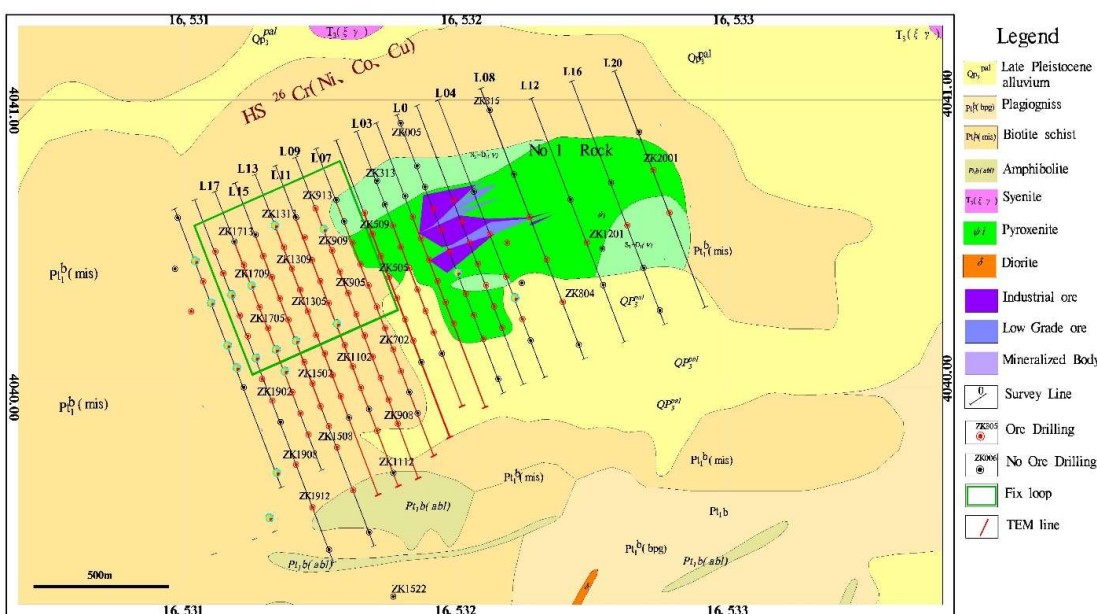

**Figure 7.** Geological map and TDEM test area.

### 3.3. Processing and Explanation

Figure 8 shows the Z-component contour and the corresponding HC zero contour with different delays. The black dotted and blue lines are the zero-contour curves of the X and Y components, respectively. According to the previous results, this corresponds to the second model, where the low resistance body underground is significantly larger than the transmitting loop.

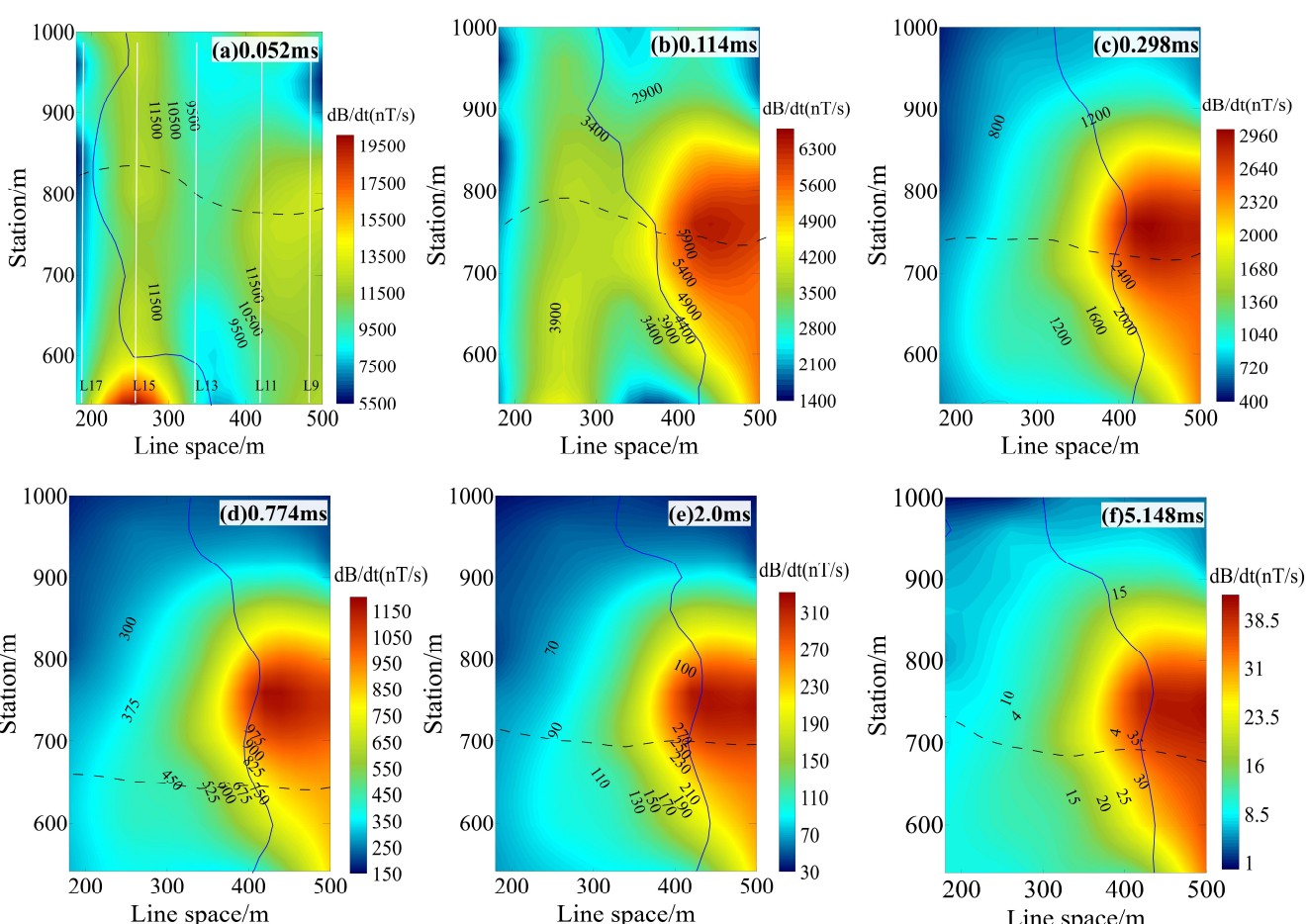

**Figure 8.** Three-component feature synthesis maps with different sampling times: (**a**) 0.052 ms; (**b**) 0.114 ms; (**c**) 0.298 ms; (**d**) 0.774 ms; (**e**) 2 ms; (**f**) 5.148 ms.

When we increased the sampling time, the Z-component anomaly center gradually shifted from the early scattered anomalies to the southeast direction. This outcome shows that the anomalous bodies underground are scattered in shallow areas, while in the deep areas, they are mainly concentrated in the southeast of the transmitter loop, and the position of the HC zero contour gradually moves to the southwest until it becomes stable. Compared with the transmitter loop's symmetry axis, the X component's zero contour obviously moves southward, and the Y component's zero contour obviously shifts to the east. This phenomenon indicates that the low-resistivity body underground has the characteristic of sloping northward and westward. According to the geological data, we found that the ore body inclines obviously to the west, and the north side is suddenly pinched out.

The intersection of the HC zero contour and the abnormal center of the vertical component point to the southeast of the transmitting loop, which indicates this is the center of a low-resistivity body underground.

### 3.4. Z-Component Quantitative Inversion

Professional TDEM software (Maxwell) was used to invert and interpret the Z-component data. Before inversion, we preprocessed the data, such as late-sampling channel interception according to noise level and appropriate filtering. According to the petrophysical measurements and borehole core logging results, the initial model of transient electromagnetic inversion is set to a layered model. One-dimensional inversion was carried out using the Beowulf module of Maxwell. Two-dimensional and three-dimensional interpolation were carried out for one-dimensional inversion data, which were displayed in two-dimensional (2D) section and three-dimensional (3D) volume, respectively, and the 2D section was compared with the existing geological section to verify the effectiveness of the method.

#### 3.4.1. Section Inversion Interpretation and Comparison

Figure 9 shows the comparison of the TDEM-inverted resistivity sections and the geological profiles of Lines 11 and 17. The geological section shows that the thickness of ore-bearing rock mass in Line 11 is larger than that of Line 17, and both are inclined in the northwest direction. The buried depth of the ore body in Line 17 is larger, and the maximum buried depth is approximately 420 m. Compared with that of Line 11, the top interface of the ore body of Line 17 is approximately 100 m lower than that of Line 11, indicating that the ore body gradually decreases from east to west. The resistivity-inversion section shows that the two survey lines have obvious northwest inclination. The thickness and buried depth of the low-resistivity area on the inversion section can objectively reflect the occurrence characteristics of the ore bodies.

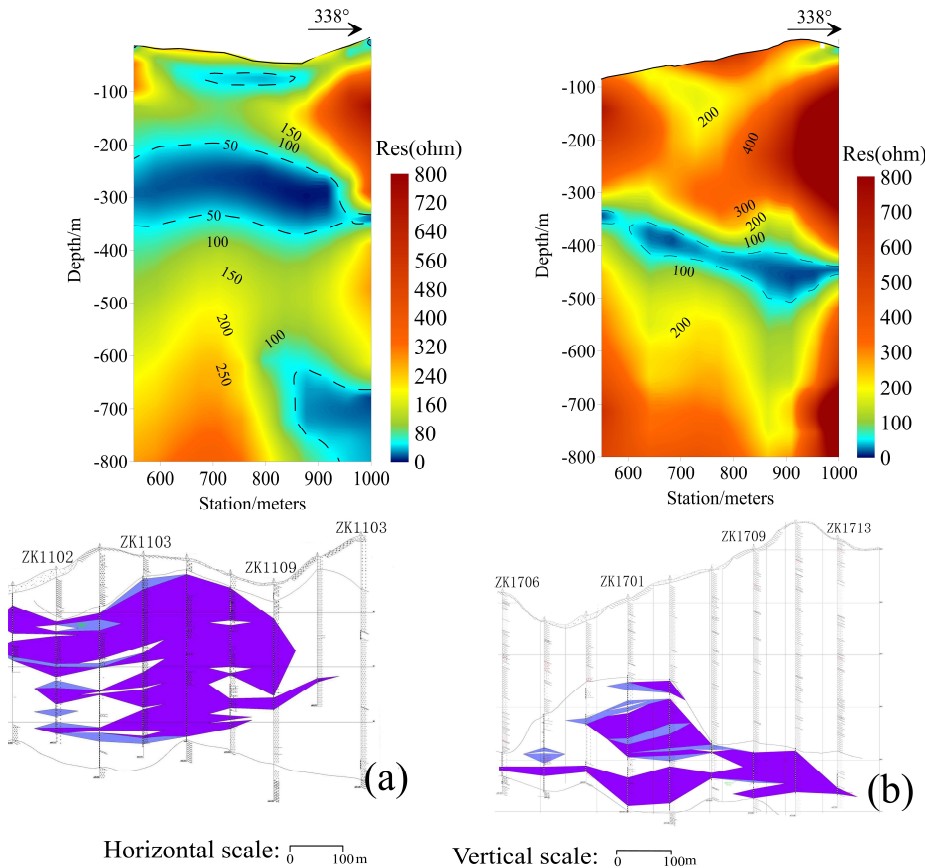

**Figure 9.** Resistivity-inversion section and geology sections for L11 and L17: (**a**) L11; (**b**) L17.

### 3.4.2. Three-Dimensional Geoelectric Model Analysis

Combined with the continuity of ore-bearing rock mass in the geological section and the TDEM-inversion results, we constructed a 3D geoelectric model in the transmitting loop. The 3D isosurface with resistivity of about 50 $\Omega\bullet$m can reflect the 3D spatial distribution characteristics of ore bodies, as shown in Figure 10. The lower part is a 3D geoelectric model, and the upper part is a 3C characteristic diagram when the sampling delay is 0.774 ms. The 3D geoelectric model has an obvious southwest tendency, and the thickest and shallowest area coincides with the high anomaly center of the vertical component and the position of the HC-zero-contour intersection. This is consistent with the forward modeling results of the second model. The qualitative interpretation results of the HC and the quantitative interpretation results of the vertical component confirm each other. The quantitative inversion results are consistent with the spatial distribution pattern of the ore-bearing rock mass, showing that the multicomponent joint interpretation is more reliable than the traditional single-component interpretation.

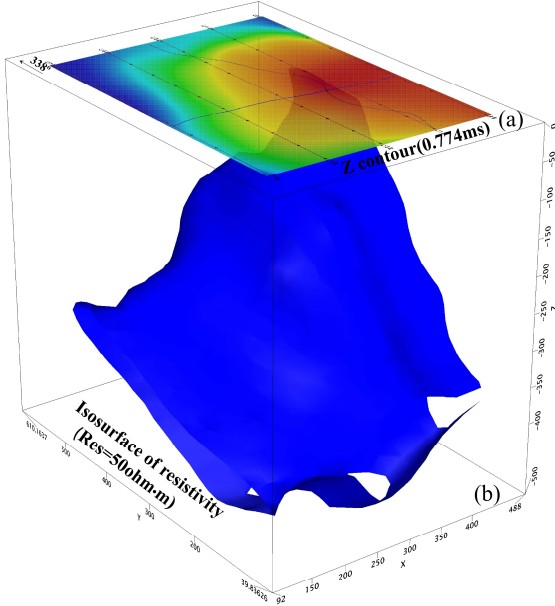

**Figure 10.** A 3D geoelectric model and multicomponent response characteristic diagram. (**a**) multicomponent response characteristic diagram at 0.774 ms; (**b**) 3D geoelectric model.

### 4. Conclusions

The terrain in the area for this study is very undulating, and the inversion work is carried out based on co-surface receiving and transmitting; there is a certain error in the inversion depth, but it does not affect the explanation of the problem. Through the inversion and interpretation of the simulated and field data, the multicomponent TDEM interpretation technique has the following advantages:

(1) Similar to the vertical component, the change trend of the zero-contour intersection of HC is directional, and its movement direction always points to the center of the ore body projected on the ground. This inspired us to quickly determine the anomaly center orientation through a 3C line in field work, which substantially reduced the cost of field data acquisition.

(2) Under certain conditions, the tendency of the ore body can be judged according to the variation characteristics of the HC 's zero contour.

(3) Multicomponent combined analysis overcomes the limitation of traditional vertical component interpretation and improves the reliability of interpretation results.

**Author Contributions:** Conceptualization: X.W. and Q.Z.; methodology: X.W., Q.Z., X.D. and J.W. (Junjie Wu); data acquisition: all participants: Y.H., J.W. (Jinghai Wang) and Q.Y.; funding acquisition: all participants; writing—review and editing: X.W. and Q.Z.; All authors have read and agreed to the published version of the manuscript.

**Funding:** This work was supported by a National Nonprofit Institute Research Grant ofIGGE (AS2022Y01).

**Data Availability Statement:** Not applicable.

**Acknowledgments:** We would like to thank Shukuan Wu engineers and Fengting Li engineers for their support and assistance during field data collection.

**Conflicts of Interest:** The authors declare no conflict of interest.

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
