# Peer review of "Multicomponent Transient Electromagnetic Exploration Technology and Its Application"

_minerals, doi:10.3390/min12060681_

Round 1
Reviewer 1 Report
It is an interesting subject. my suggestions are below
Authors should increase the fonts in all figures. Especially, geological section in fig. 8 needs vertical scale.
What kind of receiver did authors use? any explanation? (size, air cored, etc)
I am curious about the variations of 3-components. could Authors add 1 figure showing all X,Y and Z values together along a profile of their choice.
Please also add some informative details about the inversion scheme used in manuscript.
Reviewer 2 Report
Generally a good paper with a well stated objective and a field example to support the authors conclusions. I have a minor concern with the English, at times it is a bit difficult to understand the point and there is some repetition - probably some help with grammar from a native English speaker would help to tighten up the language some - this is a minor point however more clarity, I feel would make the paper more appealing. I have made some minor suggestions. Main issue is the key conclusion that adding the horizontal components gives a better interpretation than the vertical needs to be clearer for the nickel ore body case. I did not clearly understand the benefit of adding the horizontal as compared to the Maxwell inversion - as I was not sure if the Maxwell inversion used Z component only or Z and horizontal components.
